# Progressive Autoregressive Video Diffusion Models

## Abstract

*Current frontier video diffusion models have demonstrated remarkable results at generating high-quality videos. However, they can only generate short video clips, normally around 10 seconds or 240 frames, due to computation limitations during training. In this work, we show that existing models can be naturally extended to autoregressive video diffusion models without changing the architectures. Our key idea is to assign the latent frames with progressively increasing noise levels rather than a single noise level, which allows for fine-grained correspondence among the latents and large overlaps between the attention windows. Such progressive video denoising enables our models to autoregressively generate video frames without temporal inconsistency or quality degradation over time. We present the first results on text-conditioned 60-second (1440 frames) long video generation at the quality close to frontier models.*

## 1. Introduction

Frontier video diffusion models have recently demonstrated remarkable success in generating high-quality video contents [4, 10, 22, 28, 30, 33, 52] by scaling up transformer-based [31, 46] architectures. These models are capable of synthesizing realistic video sequences that are increasingly indistinguishable from real-world footage. However, despite their impressive results, current video diffusion models are constrained by a significant limitation: they can only generate videos of relatively short duration, typically up to about 10 seconds or 240 frames. This temporal restriction leads to challenges for broader applications that require longer, more continuous video outputs.

Several approaches [2, 8, 12, 15, 54] have been proposed to autoregressively apply video diffusion models for long video generation; they generate short video clips in a windowed fashion, where each subsequent clip conditions on the final frames of the previous one. One solution [8, 54] directly puts the conditioning frames into the input frames, replacing the noisy frames. Another solution [15, 41] additionally adds the same level of noise to the conditioning frames as the noisy frames. These two methods suffer from various flaws, including temporal inconsistency, abrupt scene changes, unnatural motion dynamics, and accumulated errors that lead to divergence.

In this work, we propose *progressive autoregressive video diffusion models* for high-quality long video generation. The core innovation of our method lies in the denoising process: instead of applying a single noise level across all latent frames used in traditional video diffusion models [3, 15], we progressively increase the noise levels across latent frames during denoising. Such progressive noise levels allow autoregressive video denoising with large overlaps between attention windows and fine-grained correspondence between latents with adjacent noise levels, as illustrated in Fig. 1. Our *variable length* autoregressive generation algorithm supports extending from a single latent or a chunk of latents. We additionally introduce two techniques, *chunked latents* and *overlapped conditioning*, which prevent cumulative error in autoregressive generation and frame-to-frame discontinuity. Together, our method can autoregressively generate long videos while maintaining the initial quality over time. Some parallel works [20, 37] share a similar high-level idea with our progressive video denoising, but they cannot properly extend the foundational capabilities of pre-trained video diffusion models to produce long videos at the same quality, and thus the quality and length of their results are not comparable to ours. See Appendix B for more detailed comparisons of ours and the parallel works.

Our method provides a range of benefits for the video generation community. Our method can be easily implemented by changing the noise scheduling and finetuning pre-trained video diffusion models, either UNet-based [15, 36] or Diffusion Transformer (DiT)-based [4, 29, 31] backbone, without changing the original model architecture; this allows our method to be easily reproduced and combined with orthogonal methods, such as external memory modules [12] and multiple text prompts. As shown in Sec. 4.2, our method can work training-free, if the model has been trained on varied noise levels, e.g. the masked pre-training in [54]. Moreover, the additional computational cost at inference time is minimal compared to previous works [12, 35, 47] that generate

overlapped clips, making this approach more efficient for practical use in long video generation.

We compare our method to the baselines on a text-conditioned 60-second (1440 frames) long video generation benchmark consisting of 40 real videos and their captions. Our quantitative results demonstrate that our results have overall the best quality across various dimensions and are the best at maintaining these metrics over the entire 60-second duration. Qualitatively, our method substantially outperforms the baselines in terms of temporal consistency, motion dynamics, and maintaining quality over time. Our ablation studies demonstrate the effectiveness of our *chunked latents* and *overlapped conditioning* techniques at preventing cumulative error and temporal jittering, respectively. By applying our method to two base models and outperforming their respective baselines, we confirm its universal applicability to existing video diffusion models. We encourage readers to check out our supplementary material webpage for video results for qualitatively comparing ours and the baselines.

We summarize our contribution as follows:

1. We propose a progressive noise level schedule, an autoregressive video denoising algorithm, and the chunked latents and overlapped conditioning techniques. Together, these enable high-quality long video generation building upon pre-trained video diffusion models.

2. We are the first to achieve 60-second long video generation with quality that are close to frontier models, when compared at the same resolution. On our 60-second long video generation benchmark, we achieve superior VBench and FVD scores, majority preference in human evaluations, and strong qualitative results. This marks a significant step forward in generating longer videos, a dimension that has not been explored by recent frontier video diffusion models [4, 10, 22, 28, 30, 33, 52].

3. Our method benefits the video generation research community in many ways, including easy implementation and reproduction, training-free application, minimal additional inference cost, and universal applicability on video diffusion models.

To facilitate future research, we will release our inference code based on Open-Sora [54].

## 2. Background

### 2.1. Video Diffusion Models

Diffusion models [13, 38] are generative models that learn to generate samples from a data distribution $q(\mathbf{x}^0)$ through an iterative denoising process. During training, data samples are first corrupted using the forward diffusion process $q(\mathbf{x}^t|\mathbf{x}^0)$

$$q\left(\mathbf{x}^t|\mathbf{x}^0\right) = \mathcal{N}(\mathbf{x}^t; \sqrt{\alpha^t}\mathbf{x}^0, (1-\alpha^t)\boldsymbol{I}) \quad (1)$$

$$\mathbf{x}^t = \sqrt{\alpha^t}\mathbf{x}^0 + \sqrt{1-\alpha^t}\epsilon \quad (2)$$

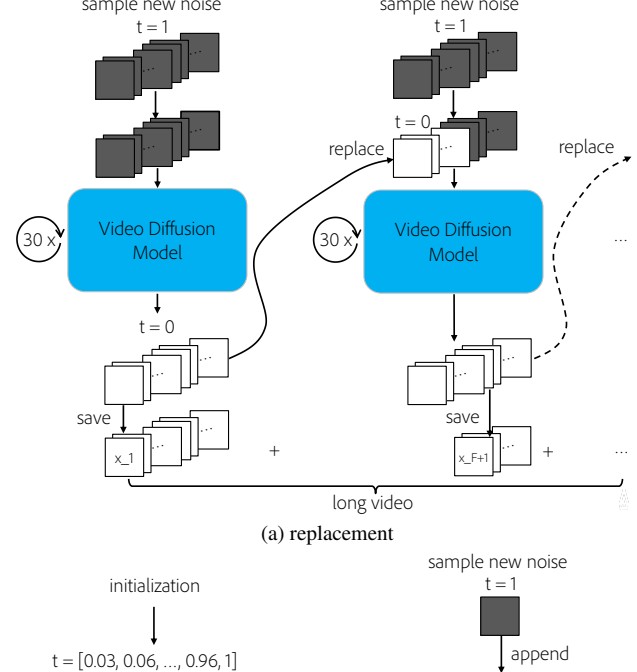

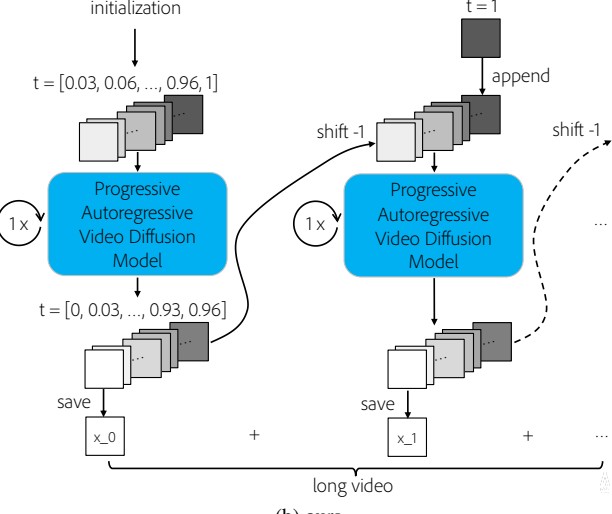

(b) ours

Figure 1. Comparison of autoregressively applying video diffusion models with *replacement* methods (top) vs. our progressive autoregressive video diffusion models (bottom). Our method allows for more fine-grained correspondence among the latent frames, where the later frames with more uncertainty can follow the pattern of the earlier, more certain frames, and larger overlap between the attention windows without extra computation cost.

where $t \in [0, T]$ is the noise level or diffusion timestep, $\epsilon \sim \mathcal{N}(\mathbf{0}, \boldsymbol{I})$ is the noise, and $\boldsymbol{\alpha}^{1:T}$ is the variance schedule. With those noisy data samples $\mathbf{x}^t$, diffusion models are trained to fit to the data distribution $q(\mathbf{x}^0)$ by maximizing the variational lower bound [21] of the log likelihood of $\mathbf{x}^0$, which can be simplified into a mean squared error loss [13]

$$\mathcal{L}(\theta) = \left\|\epsilon - \epsilon_\theta(\mathbf{x}^t, t)\right\|^2 \quad (3)$$

where $t$ is uniform between 0 and $T$, $\boldsymbol{\epsilon} \sim \mathcal{N}(\mathbf{0}, \boldsymbol{I})$ and $\boldsymbol{\epsilon}_\theta$ is the noise predicted by the model with parameters $\theta$.

At sampling time, we consider the sampling noise level schedule $\boldsymbol{\tau} = \{\tau_1, \tau_2, ..., \tau_S\}$, which is an increasing subset of $t \in [0, T]$ of length $S$ [40]. Starting from $\mathbf{x}^{\tau_S} \sim \mathcal{N}(\mathbf{0}, \boldsymbol{I}), \tau_S = T$, the reverse denoising process is autoregressively applied as

$$p_\theta\left(\mathbf{x}^{\tau_{i-1}}|\mathbf{x}^{\tau_i}\right) = q_\sigma\left(\mathbf{x}^{\tau_{i-1}}\big|\mathbf{x}^\tau, f_\theta(\mathbf{x}^t, t)\right) \qquad (4)$$

where $\hat{\mathbf{x}}^0 = f_\theta(\mathbf{x}^t, t)$ is the $\mathbf{x}^0$ predicted by the model and $f_\theta(\mathbf{x}^t, t)$ is the DDIM [40] reverse process equation, which we omit for simplicity. This gives us a sequence of samples $\mathbf{x}^T, \mathbf{x}^{\tau_{S-1}}, \ldots, \mathbf{x}^{\tau_1}, \mathbf{x}^0$, and the last sample $\mathbf{x}^0$ is the clean output result.

Video diffusion models [3, 15] are diffusion models that consider latent representations of video data, consisting of $F$ latent frames $\mathbf{x}_{0:F-1} = \{x_0, x_1, \ldots, x_{F-1}\}$. The same forward diffusion process, the reverse denoising process, and loss (Eqs. (1) to (4)) can be applied to model these video data by treating all the latents as one entity, ignoring the correlation among the latents. Recent video diffusion models [33, 54] have employed various diffusion model variants [24–26] to improve training and inference efficiency as well as output quality. Nevertheless, our method is compatible with any diffusion model variant as long as the model corrupts the data $\mathbf{x}^t$ at the same noise levels $t$.

## 2.2. Long Video Generation via Replacement

Video diffusion models can only generate short video clips, because they are only trained on videos with a limited length $F$ due to GPU memory limit. When adapted to generating $L > F$ latent frames at sampling time, their generation quality substantially degrades [35]. The straightforward solution is to autoregressively apply video diffusion models, generating each video clip while conditioning on the previous clip. In this paper, we refer to the $F$ latents that the video diffusion model processes as the *attention window*.

Given $E < F$ clean latents $\mathbf{x}_{0:E}^0$ as condition, there are two methods for autoregressively applying video diffusion models. [2, 8, 54] place the clean condition latents $\bar{\mathbf{x}}_{0:E-1}^0$ directly at the front of the attention window, directly replacing the sampled latents $\mathbf{x}_{0:E-1}^{\tau_i}$ at each denoising step

$$p_\theta\left(\bar{\mathbf{x}}_{0:E-1}^0, \mathbf{x}_{E:F-1}^{\tau_{i-1}}\big|\bar{\mathbf{x}}_{0:E-1}^0, \mathbf{x}_{E:F-1}^{\tau_i}\right) \qquad (5)$$

We will refer to this method as the *replacement-without-noise* method.

[15, 41] additionally add noise to the conditioning latents

$$p_\theta\left(\bar{\mathbf{x}}_{0:E-1}^{\tau_{i-1}}, \mathbf{x}_{E:F-1}^{\tau_{i-1}}\big|\bar{\mathbf{x}}_{0:E-1}^{\tau_i}, \mathbf{x}_{E:F-1}^{\tau_i}\right) \qquad (6)$$

where $\bar{\mathbf{x}}_{0:E-1}^{\tau_i}$ are the condition latents $\bar{\mathbf{x}}_{0:E-1}^0$ noised via the forward process (Eqs. (1) and (2)). This maintains the

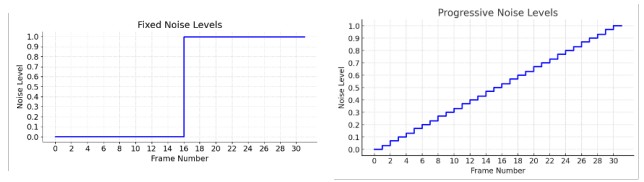

Figure 2. Comparison of noise levels of the replacement without noise method (left) vs. ours (right).

same noise level distribution and training objective as regular video diffusion models. We will refer to this method as the *replacement-with-noise* method. Note that [15] proposes reconstruction guidance for the *replacement-with-noise* method but is not widely adopted.

Both the *replacement-with-noise* method and the *replacement-without-noise* method allow a video diffusion model to autoregressively generate video frames by conditioning on previous frames. We consider them as baselines in our experiments in Sec. 4.2.

See Appendix B for a detailed discussion of a few concurrent works [5, 20, 37] that share a high-level idea similar to our work. Please refer to Appendix C for related works.

## 3. Progressive Autoregressive Video Diffusion Models

In this work, we consider long video generation with video diffusion models. Existing video diffusion models [1, 2, 11, 33] can only generate short video clips up to a limited length $F$ (e.g. 10 seconds or 240 frames), due to GPU memory constraints during training. We show that, without changing the architectures, they can be naturally extended to *progressive autoregressive* video diffusion models that can generate long videos without quality degradation. We achieve this by proposing a per-frame noise schedule, which is inspired by [5]. During training, we finetune pre-trained video diffusion models to adapt to such noise schedule; during sampling, our models adopt such noise schedule and can thus autoregressively generate video frames.

### 3.1. Progressive Video Denoising

Conventional video diffusion methods assign a single noise level to all the latent frames. Inspired by [5], we assign progressively increasing noise levels $\tau_{0:F-1}$ to the $F$ latent frames in the attention window. These noise levels can be obtained from the sampling noise level schedule $\boldsymbol{\tau} = \{\tau_0, \tau_1, ..., \tau_S\}$, where $0 = \tau_0 < \ldots < \tau_{i-1} < \tau_i < \ldots < \tau_S = T$, depending on $F$ and $S$. In this work, we consider the linear sampling schedule

$$\boldsymbol{\tau}'_{0:S} = \left\{\tau'_0, \tau'_1, \tau'_2, \ldots, \tau'_{S-1}, \tau'_S\right\}$$
$$= \left\{0, \frac{T}{S}, \frac{2T}{S}, \ldots, \frac{(S-1)T}{S}, T\right\} \qquad (7)$$

---

**Algorithm 1** Inference procedure of progressive autoregressive video diffusion models

---

**Require:** Initial video latent frames $\mathbf{x}_{0:F-1}^0 = \{\mathbf{x}_0^0, \mathbf{x}_1^0, ..., \mathbf{x}_{F-1}^0\}$, maximum noise level $T$, number of inference steps $S$, and attention window size $F = S$

1: $\boldsymbol{\tau}_{0:S}' = \{\tau_0', \tau_1', \ldots, \tau_S'\} = \left\{0, \dfrac{T}{S}, \ldots, T\right\}$      $\triangleright$ Eq. (7), progressive noise levels

2: $\boldsymbol{\epsilon} \sim \mathcal{N}(\mathbf{0}, \boldsymbol{I})$

3: $\mathbf{x}_{0:F-1}^{\boldsymbol{\tau}_{1:S}'} = \sqrt{\alpha^{\boldsymbol{\tau}_{1:S}'}}\mathbf{x}_{0:F-1}^0 + \sqrt{1 - \alpha^{\boldsymbol{\tau}_{1:S}'}}\boldsymbol{\epsilon}$      $\triangleright$ Eq. (2), add noise

4: **for** each autoregressive generation step $i = 1, 2, \ldots, N$ **do**

5:     $\mathbf{x}_{0:F-1}^{\boldsymbol{\tau}_{0:S-1}'} = \left\{\mathbf{x}_0^0, \mathbf{x}_1^{\tau_1'}, \ldots, \mathbf{x}_{F-1}^{\tau_{S-1}'}\right\} \sim p_\theta\left(\mathbf{x}_{0:F-1}^{\boldsymbol{\tau}_{0:S-1}'}\Big|\mathbf{x}_{0:F-1}^{\boldsymbol{\tau}_{1:S}'}\right)$      $\triangleright$ Eq. (8), one sampling step

6:     $\mathbf{x}_{F-1}^T \sim \mathcal{N}(\mathbf{0}, \boldsymbol{I})$      $\triangleright$ Sample a new noisy latent

7:     Append $\mathbf{x}_0^0$ to the list of clean latents

8:     $\mathbf{x}_{0:F-1}^{\boldsymbol{\tau}_{0:S}'} = \left\{\mathbf{x}_1^{\tau_1'}, \ldots, \mathbf{x}_{F-2}^{\tau_{S-1}'}, \mathbf{x}_{F-1}^T\right\}$      $\triangleright$ Remove $\mathbf{x}_0^0$, shift latents by $-1$, and append $\mathbf{x}_{F-1}^T$

9: **end for**

10: **return** List of clean frames

---

For simplicity, we set $F = S$, so that our latents' noise levels $\boldsymbol{\tau}_{0:F-1}$ can take the values of either $\boldsymbol{\tau}_{0:S-1}'$ or $\boldsymbol{\tau}_{1:S}'$. We denoise the latent frames of a video with the progressively increasing noise levels

$$p_\theta\left(\mathbf{x}_0^{\tau_0'}, \mathbf{x}_1^{\tau_1'}, ..., \mathbf{x}_{F-2}^{\tau_{S-2}'}, \mathbf{x}_{F-1}^{\tau_{S-1}'}\Big|\mathbf{x}_0^{\tau_1'}, \mathbf{x}_1^{\tau_2'}, ..., \mathbf{x}_{F-2}^{\tau_{S-1}'}, \mathbf{x}_{F-1}^{\tau_S'}\right) \tag{8}$$

where the input frames and output frames have noise levels $\boldsymbol{\tau}_{1:S}'$ and $\boldsymbol{\tau}_{0:S-1}'$ respectively.

With our progressive noise levels, which is a subset of independent noise levels $\mathbf{t}_{0:F-1}$ for the frames $\mathbf{x}_{0:F-1}$, one concern is that whether this formulation is still compatible with existing video diffusion models. Conceptually, we are effectively using a single set of model parameters $\theta$ to jointly model $F$ latents $q(\mathbf{x}_0^0), q(\mathbf{x}_1^0), \ldots, q(\mathbf{x}_{F-1}^0)$, where each latent has a single noise level $t_f$ like regular diffusion models. Thus, we can directly apply the forward process, reverse process, and training loss (Eqs. (1) to (4)) of existing video diffusion models [13, 25, 26, 40]. This means that we can obtain our progressive autoregressive video diffusion models from pre-trained video diffusion models, which demands immense computation resources.

Intuitively, the benefit of our progressive video denoising process is that it gradually establishes correlation among consecutive latent frames. Given some existing video latents as conditioning, it is challenging for video diffusion models to produce temporally consistent extensions latents from newly sampled noisy latents [35]. In contrast to the *replacement-with-noise* method [2, 15] where the latents are denoised together at the same noise level, our progressive video denoising encourages the later latents with higher uncertainty to follow the patterns of the earlier and more certain latents, facilitating modeling a smoother temporal transition and better preserving motion velocity. Compared to the

*replacement-without-noise* method where there is a large noise level gap between the clean condition latents $\bar{\mathbf{x}}_{0:E-1}^0$ and the noisy frames $\mathbf{x}_{E:F-1}^{\tau_i}$, our method provides more fine-grained correspondence, where the difference between neighboring noise levels is only $\frac{T}{S}$, as illustrated in Eq. (7) and Fig. 2.

## 3.2. Autoregressive Generation

Notice that the noise levels of the input and output latents in Eqs. (7) and (8), $\boldsymbol{\tau}_{0:S-1}'$ and $\boldsymbol{\tau}_{1:S}'$, only differ by $\tau_0' = 0$ and $\tau_S' = T$. We can simply transition the output latents into the correct input noise levels by saving and removing the clean latent $\mathbf{x}_0^0$ at the front, shifting the latent sequence forward once, and appending a new noisy latent $\mathbf{x}_{F-1}^T \sim \mathcal{N}(\mathbf{0}, \boldsymbol{I})$ at back, as detailed in Alg. 1 and Fig. 1.

**Variable Length** The above design only allows for autoregressive video extension given an initial video of length $F$. In addition, the noisy latents remaining in the attention window $\mathbf{x}_{0:F-1}^{\boldsymbol{\tau}_{1:S}'}$ (line 8 of Alg. 1) are discarded after the end of the autoregressive inference, which can cause wasted computing resources and inaccurate handling of the ending of text prompt. To enable text-to-long-video generation without starting latents and proper generation of ending latents, we extend the base design in Eq. (8) and Alg. 1 to add an initialization stage and an termination stage, where the model operates on variable attention window lengths from 1 to $F - 1$. During initialization, we simply disable the "removing $\mathbf{x}_0^0$" operation in line 8 of Alg. 1: starting from a noisy latent $\{\mathbf{x}_0^T\}$, we denoise and append to obtain $\{\mathbf{x}_0^{\tau_{S-1}'}, \mathbf{x}_1^T\}$; we repeat this by $F - 1$ times to obtain $\mathbf{x}_{0:F-1}^{\boldsymbol{\tau}_{1:S}'} = \left\{\mathbf{x}_1^{\tau_1'}, \ldots, \mathbf{x}_{F-2}^{\tau_{S-1}'}, \mathbf{x}_{F-1}^T\right\}$, i.e. the input to line 5 of Alg. 1. During termination, we disable the "append $\mathbf{x}_{F-1}^T$" operation in line 6 and 7 of Alg. 1: starting with $F$ latents

$\mathbf{x}_{0:F-1}^{\boldsymbol{\tau}'_{1:S}} = \left\{ x_0^{\tau'_1}, \ldots, x_{F-2}^{\tau'_{S-1}}, x_{F-1}^T \right\}$, we denoise, save and remove to obtain $\mathbf{x}_{0:F-2}^{\boldsymbol{\tau}'_{1:S-1}} = \left\{ x_0^{\tau'_1}, \ldots, x_{F-2}^{\tau'_{S-1}} \right\}$; we repeat this by $F$ times to save and remove all the remaining latents in the attention window. We train the model accordingly on video latent frames with variable lengths ranging from $1$ to $F-1$, following the noise levels described above.

### 3.3. Chunked Latents

3D VAEs [21, 33, 54] usually encode and decode video latent frames chunk-by-chunk. In our early experiments, we find that naively implementing our method on video diffusion models, i.e. when all latents are given different noise levels and the attention window is shifted by one latent at a time, leads to serious cumulative error and the videos diverge quickly after a few seconds, as shown in Ablation 2 in Fig. 6. We resolve the problem by *treating a chunk of latents as a whole*: they are assigned with the same noise level, and are added and removed from the attention window together. In other words, for a 3D VAE chunk size of $C$ latents, e.g. $C = 5$ mentioned in our training details in Sec. 4, we shift the attention window by $C$ latents every $C$ sampling steps. Effectively, the $C$ latents that belong to the same chunk always have the same noise level $t$ and are added to or removed from the attention window together. Our ablation experiments shows that, for models using a 3D VAE, treating a chunk of latents as a whole effectively prevents accumulated errors that would lead to divergence.

### 3.4. Overlapped Conditioning

In our early experiments, naively implementing our method on video diffusion models results in temporal jittering. We hypothesize that this is because the clean latents $\mathbf{x}_{0:C-1}^0$ are immediately removed from the attention window; as the later latents cannot attend to the previous clean latents, it is hard for the model to denoise the later latents to be perfectly temporally consistent with the previous clean latents. In practice, we always keep a chunk of $C$ clean latents in front of all the noisy latents, extending the attention window lengths $F$ by $C$. Our ablation study shows that overlapped conditioning helps resolving the frame-to-frame discontinuity issue.

*Overlapped conditioning* requires an additional inference cost at $C/F$ (5/50 in our implementation) of the original cost. When using the same number of conditioning frames $E$ and $F$, the replacement methods [8, 15, 54] and ours have the same inference efficiency. The key advantage of our method is that the large overlap of noisy frames enables the model to preserve the high-level information—such as motion—from prior frames. Thus, we only need a single chunk of $C$ clean condition frames to propagate high-frequency details and prevent per-chunk temporal jittering. In contrast, the replacement methods need to balance the tradeoff between more

overlap between video clips or better inference efficiency. In practice, their implementation [54] often use one chunk of frames as condition to save inference computation, but the limited overlap causes unnatural motion transition and abrupt scene changes across clips, as discussed in Sec. 4.2.

### 3.5. Training

While regular video diffusion models always treat the latent frames as a single entity by assigning a single noise level $t$ to all of them, we extend this formulation by modeling each frame independently and assigning them with varying noise levels $\mathbf{t}_{0:F-1}$. This change in the noise level distribution typically requires finetuning the pre-trained video diffusion model to adapt to our progressive noise level distribution. We finetune pre-trained video diffusion models by modifying the noise levels during training. Regular diffusion model training [13, 25, 26] involves uniformly sampling a noise level $t \in [0, T)$, adding noise to the samples $\mathbf{x}_{0:F-1}^0$ via the forward diffusion process (Eqs. (1) and (2)), and computing the loss (Eq. (3)). To enable sampling with progressive noise levels in Eqs. (7) and (8), we simply switch to per-frame training noise levels $\mathbf{t}_{0:F-1}$. In our experiment, we observed that, similar to the sampling noise levels $\boldsymbol{\tau}'_{0:S}$ in Eq. (7), training on a simple linear noise schedule yielded satisfactory results for all reported experiments. During training, the noise levels $\boldsymbol{t}$ is perturbated by a random shift $\delta$ to preserve the coverage of the full diffusion timestep range $[0, T)$ [39]. $\delta = 0.4\epsilon(t_i - t_{i+1}), \epsilon \sim \mathcal{N}(0, \boldsymbol{I})$ is randomly sampled for each training iteration and remains constant for all $\mathbf{t}_{0:F-1}$ within that iteration.

## 4. Experiments

In this section, we follow our terminology as discussed and defined in Sec. 2.2, including the *replacement-with-noise* method, the *replacement-without-noise* method, and attention window.

### 4.1. Implementation

**Our models and baseline models** We implement our progressive autoregressive video diffusion models by finetuning from pre-trained models. Specifically, we use two video diffusion models based on the diffusion transformer architecture [4, 31]: Open-Sora v1.2 [54] (denoted as $O$) and a modified variant of Open-Sora (denoted as $M$ in later experiments). Both models are latent video diffusion models [3], each utilizing a corresponding 3D VAE that encodes 17 ($O$) or 16 ($M$) video frames into 5 latent representations. $O$ generates videos at 240×424 resolution 24 FPS with 30 sampling steps. $M$ produces results at 176×320 resolution 24 FPS with 50 sampling steps. Based on $O$ and $M$, we also implement two baseline autoregressive video generation methods, *replacement-with-noise* (denoted as RW) and *replacement-without-noise* (denoted as RN) (Sec. 2.2), to

Table 1. Quantitative comparison of our progressive autoregressive video generation (PA) and two baseline methods *replacement-with-noise* (RW) and *replacement-without-noise* (RN) on two base models (*M* and *O*), and other baselines StreamingT2V [12], Stable Video Diffusion (SVD) [2], and FIFO-Diffusion [20].

| | Subject Consistency ↑ | Background Consistency ↑ | Motion Smoothness ↑ | Dynamic Degree ↑ | Aesthetic Quality ↑ | Imaging Quality ↑ | Num Scenes ↓ | FVD ↓ |
|---|---|---|---|---|---|---|---|---|
| **PA-*M*** (ours) | 0.7923 | **0.8964** | 0.9896 | 0.8000 | **0.4726** | **0.5927** | 1.75 | **358.020** |
| RW-*M* | 0.8001 | 0.8851 | 0.9836 | 0.3958 | 0.4123 | **0.5961** | 1.10 | 669.747 |
| **PA-*O*-base** (ours) | 0.7656 | 0.8880 | 0.9859 | 0.5625 | 0.4582 | 0.5033 | 2.04 | 548.117 |
| RN-*O*-base | 0.7406 | 0.8820 | 0.9873 | 0.5750 | 0.4034 | 0.4464 | 5.19 | 600.690 |
| StreamingSVD | **0.8172** | 0.8916 | **0.9929** | 0.65 | 0.4264 | 0.5566 | **1.08** | 440.272 |
| SVD-XT | 0.6102 | 0.8136 | 0.9724 | **0.9875** | 0.3019 | 0.4814 | 2.10 | 702.343 |
| FIFO-OSP | 0.7577 | 0.8990 | 0.9731 | 0.75 | N/A | 0.5675 | 18.32 | 975.459 |

compare with our proposed *progressive autoregressive* (denoted as PA) video generation method (Sec. 3).

We train *M* on our progressive noise levels, as discussed in Sec. 3.5. The resulting model can perform progressive autoregressive video generation, which we denote as PA-*M*. We also train *M* with the *replacement-with-noise* method, which we will denote as RW-*M*. Starting from the same pretrained weight of the base model, RW-*M* is trained for 3 times more training steps compared to PA-*M*.

*O* undergoes masked pre-training [54], where the masked latents $\mathbf{x}_{0:E-1}^0$ are clean without any added noise [54]. This allows the *O* base model to perform autoregressive video generation with the *replacement-without-noise* method. We denote this model as RN-*O*-base. Such training also allows *O* to learn that the noise levels $\mathbf{t}_{0:F-1}$ can be independent with respect to the latent frames and thus enables our *progressive autoregressive* video denoising sampling procedure (Sec. 3.2 and Alg. 1) to work training-free. We denote this model as PA-*O*-base. Please refer to Appendix E for training details.

## 4.2. Long video generation

**Baselines** As discussed in Sec. 4, using our base models, we implement two baseline autoregressive video generation methods on three models, which are denoted as RW-*M*, RN-*O*-base, and RN-*O*. We also compare to Stable Video Diffusion (SVD) [2] and StreamingT2V [12] model families. Specifically, we consider the SVD-XT model from SVD, a image-to-video model that generates a short video clip of 25 frames at 576x1024 resolution given an conditioning image. We apply it autoregressively, using the last image of the previous clip as the condition for generating a new clip. This is equivalent to the *replacement-without-noise* method except that it only conditions on a single frame rather than a chunk of 17 frames as RN-*O*. We also consider the StreamingSVD model from StreamingT2V, a image-to-long-video generation model that uses SVD as the base

model [12]; its autoregressive video generation is enabled by training additional modules that connect to the base model via cross-attention. Similar to our progressive autoregressive video diffusion models, StreamingSVD can autoregressively generate long videos at 720x1280 resolution with arbitrary lengths, which we set to 1440 frames. We also compare to a concurrent work FIFO-Diffusion [20] implemented on Open-Sora-Plan v1.0.0 [23], denoted as FIFO-OSP. It generates at 256x256 resolution with a context window of 65 latent frames. See Appendix B for a discussion on [20] and other concurrent works. See Appendix F for details on our testing set, quantitative metrics, and traditional video quality evaluation.

**Metrics** We consider 6 metrics in VBench [18]: subject consistency, background consistency, motion smoothness, dynamic degree, aesthetic quality, and imaging quality. We compute average metrics using VBench-long, where each metric is computed on 30 2-second clips for each 60-second video; for subject and background consistency, a clip-to-clip metric is considered in addition to the average metric over the clips. We also show how the metrics vary over time by plotting the metrics over the 30 2-second clips averaged over the 80 60-second videos.

Similar to [12], we also use the Adaptive Detector algorithm from PySceneDetect [34] to count the number of detected scene changes, where Num Scenes = 1 means that there is no scene change detected.

We also compute Fréchet Video Distance (FVD) [45] to measure the overall quality of the generated videos compared to real videos. We adopt the improved implementation of FVD proposed in [9] using the VideoMAE-v2 [48] model. The FVD metric usually requires a large number of video samples in order to produce a reliable value. Since our testing set includes only 40 real videos and each model only generate 80 videos, naively computing FVD on them results in erroneous values such as -3.62e+64. Instead, we compute

FVD on the 2-second clips of the long videos, so that we have 1495 real videos and 2400 generated videos.

**Quantitative Results** We present the average metrics for each model in Tab. 1. The metrics are averaged over all the videos that each model generates from our testing set described above. Our PA-*M* has the best results overall. Notably, it surpasses other methods in FVD by a substantial margin, illustrating that its results are the most realistic. It also achieves either the best or close-to-best in other metrics. Its *replacement-with-noise* counterpart, RW-*M*, suffers from poor Dynamic Degree and FVD, because its videos are mostly static. Our RW-*O*-base surpasses its *replacement-without-noise* counterpart RN-*O*-base in all metrics except for being close at Dynamic Degree, while using the exact same model parameters without any finetuning. RN-*O*-base mainly suffers from a high number of scene changes.

In Fig. 3, we also illustrate the trend of metrics over the 1-minute duration of videos for each model. Our models *M*-PA and *O*-PA can best maintain the level of all metrics, while their *replacement*-method counterparts, *M*-RW and *O*-RN, both exhibit distinct reduction in dynamic degree, aesthetic quality, and imaging quality.

**Qualitative Results** We also show strength of our method with qualitative comparison results in Fig. 5. Both of our models demonstrate strong performance in terms of frame fidelity and motion realism (e.g. camera motion, wave motion, and running gestures) and outperforms other baselines. For more qualitative results, please refer to our supplementary material webpage.

**User study** We conduct a human evaluation with 12 users to compare the generated videos from each method. As shown in Fig. 4, our PA-*M* is favored in each duel by a large margin.

### 4.3. Ablation Study

We conduct ablation studies on the PA-*M* model to evaluate the impact of chunked latents (Sec. 3.3), and overlapped conditioning (Sec. 3.4). Qualitative comparison is shown in Fig. 6 and in the supplementary material webpage. In Ablation 1, we observe that the absence of clean frames in the input sequence prevents noisy frames from attending to previous clean frames, resulting in poor performance over a long duration. This also causes frame-to-frame discontinuity, which is more noticeable in the supplementary anonymous webpage. In Ablation 2, not decoding the video chunk-by-chunk leads to severe cumulative errors, causing the video to diverge after only a few seconds.

See Appendix H for additional ablation study on variable length and the number of sampling steps $S$.

### 5. Conclusion

In this work, we target long video generation, a fundamental challenge of current video diffusion models. We

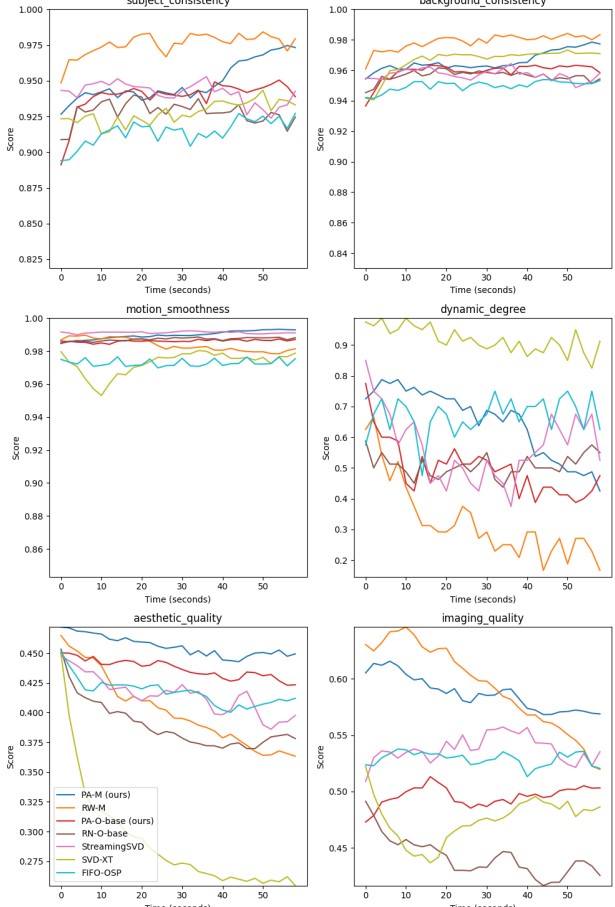

Figure 3. VBench [18] scores over the 60-second duration, which are computed on 30 2-second clips.

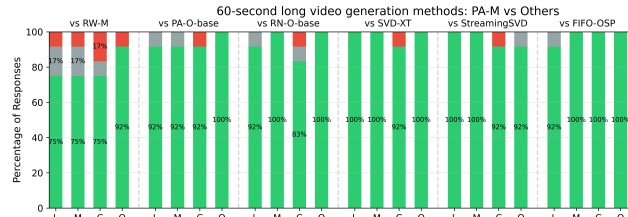

Figure 4. Human evaluation results comparing long video methods on long-shot (L), motion (M), temporal consistency (C), and overall (O).

show that they can be naturally adapted to become progressive autoregressive video diffusion models without changing the architectures. With our progressive noise levels and the autoregressive video denoising process (Secs. 3.1 and 3.2), we obtain state-of-the-art results on long video generation at 1-minute long. Since our method does not require model architecture changes, it can be seamlessly combined with orthogonal works, paving the way for generating longer videos at higher quality, long-term dependency, and controllability.

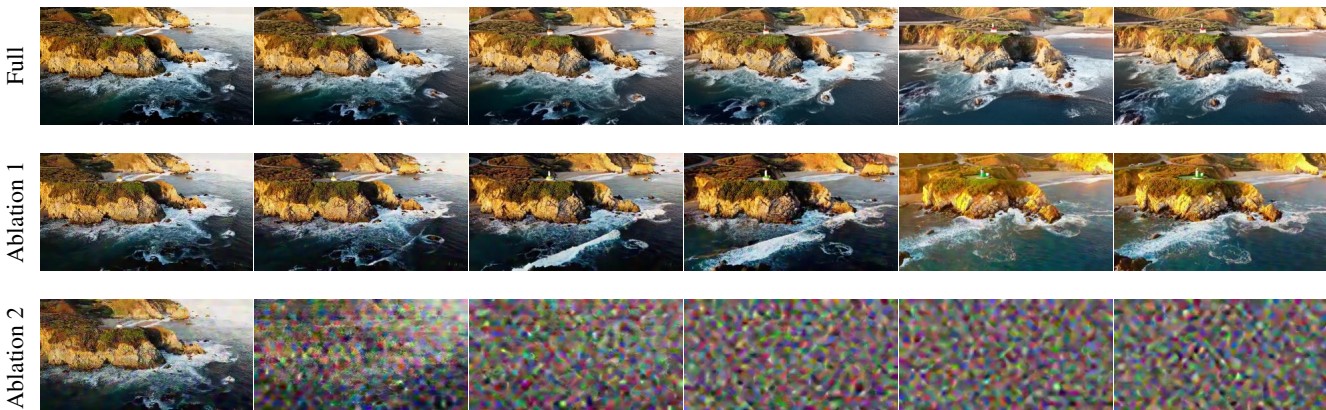

Figure 5. Qualitative comparison of PA-*M* (ours), RW-*M*, PA-*O*-base (ours), RN-*O*-base, StreamingSVD from StreamingT2V [12], SVD-XT from Stable Video Diffusion [2], and FIFO-Diffusion [20]. Frames are evenly sampled from 1 minute long generated video, i.e. at 10, 20, 30, 40, 50, and 60 seconds. Our models can autoregressively generate 60-second, 1440-frame videos without quality degradation.

Figure 6. Qualitative comparison for ablation study. Full represents for our full solution based on PA-*M*, Ablation 1 is with *chunked latents* but without *overlapped conditioning*. Ablation 2 is without both techniques. The frames are evenly sampled from 16-second generated videos.

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
