# OpenReview forum: "Progressive Autoregressive Video Diffusion Models"
_thecvf.com/CVPR/2025/Workshop/CVEU — CVPR 2025_

### Official Review · Reviewer_o3Cr · 2025-03-13

**Rating:** 4
**Confidence:** 3

**Review:**

Summary:

This paper proposes an autoregressive video generation model based on diffusion models. To adapt diffusion models to autoregressive generation, the authors assign progressively increasing noise levels to latent frames. In addition, two techniques called chunked latents and overlapped conditioning are introduced to mitigate error accumulation and temporal jitter, respectively. The effectiveness of the proposed model is verified through quantitative comparisons on a 60-second video generation benchmark.

Strengths:

* The idea of progressively increasing noise levels is very intuitive, it can be seen as a soft-masked version of sliding windows.
* The qualitative results for 60-second video generation are quite good, with much less degradation over time compared to the other baselines. The quality score evolution in Figure 3 is straightforward and clearly demonstrates the advantages of the proposed method.

Weaknesses:

* Lack of empirical comparison with Diffusion Forcing [1]. As mentioned in the supplementary material, the authors noise history latents with progressively increasing noise levels, while Diffusion Forcing noises them with independent uniform noise levels. It would be interesting to compare the two strategies and see if there are advantages of each.
* The chuncked latent design is not directly compatible with more recent VAEs that use a temporal tiling strategy, which linearly blends VAE latents in the overlapping regions. This is commonly used in Meta Movie Gen [2] and other recent work.
* The overlapped conditioning design seems to be unfriendly to the KV cache, limiting the inference efficiency. The authors could further compare the inference cost of different models, which would be important for future real-time autoregressive generation applications.

---

[1] Chen, et al. Diffusion Forcing: Next-token Prediction Meets Full-Sequence Diffusion. NeurIPS 2024.

[2] Meta. Movie Gen: A Cast of Media Foundation Models. 2024.

---

### Official Review · Reviewer_zc9G · 2025-03-22

**Rating:** 4
**Confidence:** 4

**Review:**

This paper introduces progressive autoregressive video diffusion models for long video generation. Instead of a single noise level for all frames, this work progressively increases the noise levels across frames during denoising.

Strengths:
- Important task: Generating long videos that go beyond the training length is important, since it is not clear if in the nearby future long video generation will be solved just with larger models trained on very long videos.
- Code release: This approach should be applicable to most video models and the provided code will help reproducing the results and build upon it.

Weaknesses:
- Only one backbone: It would have been great to test the idea on multiple backbones, not only OpenSora. Especially, because OpenSora 1.2 doesn’t use 3D attention like all other video DiT models.
- Visual results: While the visuals look better than previous works in long generation, they are generally low quality because of the outdated video base model. Also there are some cut artifacts visible.

The paper is decent and the results look promising, though they could be more polished with more recent video models. For a workshop this is enough, even if more backbones would have been great to demonstrate generalizability.

---

### Official Review · Reviewer_ZgPj · 2025-03-25
**A novel noise schedule to auto-regressively generate longer video sequences**

**Rating:** 4
**Confidence:** 4

**Review:**

This paper proposes a new noise schedule that progressively increases noise for later frames in an auto-regressive video generation framework.

The motivation is solid, and the concept is clearly illustrated in Figures 1 and 2.

Details of the progressive auto-regressive video diffusion approach are explained in Section 3.

Both qualitative and quantitative results demonstrate improvements.

Although the approach is simple, it is effective and well-motivated.

Therefore, I recommend a weak accept.

---

### Decision · Program_Chairs · 2025-03-25

**Decision:**

Accept

**Comment:**

The paper introduces a progressive noise schedule for autoregressive video generation using diffusion models. Reviewers praised the clarity and intuition behind the method, solid motivation, and clear empirical improvements over baselines. Though some limitations were noted—such as reliance on a single backbone, lack of comparisons with recent related methods, and limited exploration of inference efficiency—overall, the contributions are meaningful and suitable for the workshop.

Given the positive assessments of motivation, method clarity, and promising results, the paper is clearly accepted. Authors are encouraged to include additional comparisons, address generalizability with other backbones, and clarify inference efficiency considerations in the camera-ready version.